# Long-Term Performance of a Hybrid-Flow Constructed Wetlands System for Urban Wastewater Treatment in Caldera de Tirajana (Santa Lucía, Gran Canaria, Spain)

**DOI:** 10.3390/ijerph192214871

**Published:** 2022-11-11

**Authors:** Gilberto M. Martel-Rodríguez, Vanessa Millán-Gabet, Carlos A. Mendieta-Pino, Eva García-Romero, José R. Sánchez-Ramírez

**Affiliations:** 1Water Department, Instituto Tecnológico de Canarias (ITC), 35119 Santa Lucía, Spain; 2Department of Process Engineering, University of Las Palmas de Gran Canaria (ULPGC), 35214 Las Palmas de Gran Canaria, Spain; 3Mancomunidad Intermunicipal del Sureste de Gran Canaria, 35118 Agüimes, Spain

**Keywords:** hybrid vertical- and horizontal-flow constructed wetlands, municipal wastewater treatment, small and remote community, water quality parameters, pollutant removal efficiency, 11 years’ experience, 6 years monitoring

## Abstract

This paper describes the results that have been obtained in a real case study of a hybrid constructed wetlands system, which has been in continuous operation for over 11 years. The main aim of the study was to understand the long-term operation and efficiency of the system (which is situated in the municipality of Santa Lucía, Gran Canaria, Spain), which comprises two vertical-flow and one horizontal-flow constructed wetlands for the treatment of urban wastewater. The system, which was originally designed to treat a flow rate of 12.5 m^3^/day, with a load of 100 equivalent inhabitants, has been operating since its inauguration (July 2008), with a flow rate of almost 35 m^3^/day and a load of 400 equivalent inhabitants. Despite this, the mean total removal efficiencies during the study period (2014–2019) are optimal for a system of these characteristics, as follows: 92% for 5-day biochemical oxygen demand (BOD_5_), 89% for the chemical oxygen demand (COD), and 97% for the total suspended solids (TSS). The system efficiency, with respect to nutrient removal, was somewhat lower, resulting in 48% for total N and 35% for NH_4_. It has been confirmed with this study that this type of system is an appropriate, robust, resilient nature-based solution for the treatment of the wastewater that is generated in small communities, especially in zones with a warm climate, stable mean temperatures, and mild winters.

## 1. Introduction

The Canary Islands (Spain) are particularly vulnerable to the effects and the consequences of climate change. As well as the already scarce availability of fresh water in the islands and its negative impact on the natural water cycle, the demand for water is rapidly growing due to a rising population and standard of living, an increasing number of tourist visitors, the need for more locally produced food, greater irrigation requirements, etc.

In consequence of the above issues, the water deficit situation in the archipelago is worsening. While this is partially, and artificially, alleviated through desalination technologies, the processes that have been employed bring with them other issues that need to be resolved, including high energy consumption and CO_2_ emissions, the emission of reject brine into the sea, etc.

Safe reclaimed water reuse is one of the non-conventional solutions that are being considered around the world as an effective and necessary alternative source of water in order to alleviate climate-change-derived water scarcity problems, compensating for the water deficit that is associated with droughts and water crises [1,2,3], especially in the agriculture sector [4], which is the main beneficiary of the reclaimed water in islands such as Gran Canaria.

Both the production and the treatment of water have a direct association with energy [5]. A wide variety of technologies that can be adapted to different requirements and environments must be available. Similarly, the tools that are needed in order to enable decision making based on economic, environmental, and social criteria must also be at hand. The energy consumption that is associated with water production and management generates large amounts of greenhouse gases (GHG), which contribute to climate change [6].

The appropriate final disposal of the wastewater that is generated by populated settlements and their different associated economic activities, as well as its safe recycling and reuse, has become a growing challenge at a global scale, which has been acknowledged as such by the United Nations in its Sustainable Development Goal (SDG) 6: clean water and sanitation for all [7].

SDG 6 widens the focus of the Millennium Development Goals (MDGs) for potable water and basic sanitation in order to cover the whole water cycle, including water management, wastewater, and ecosystem resources [8].

In large urban agglomerations, the most widespread solution for this problem is the transportation of effluents through networks of sewage pipes to wastewater treatment plants (WWTPs), where most of the pollutants that are present in the water are removed, allowing it to be discharged into the public domain or sent to regenerated water networks for its reuse.

This model requires suitable treatment technologies and the availability of sufficient electrical energy in order to guarantee efficient and appropriate treatment in the smallest space possible; however, it has the major disadvantage, apart being very costly [9], of its associated GHG emissions [10,11]. Such technology- and energy-related conditions are, however, not always possible when dealing with the small-scale treatment and reuse of wastewater, whether in rural environments or in developing countries. This circumstance opens up the possibility of the use of technologies that are less energy intensive and, in consequence, are associated with a lower overall climate change impact [10,12].

The European Directive 91/271/EEC [13] on urban wastewater treatment sets out the requirements for the discharge of treated water from wastewater treatment installations. Different treatment stages have to be applied in order to ensure that the required values for the different parameters that are included in the directive are met. It should be noted here that the same purification technologies cannot be applied in the design of WWTPs for large urban settlements as those for small communities [14]. Moreover, in the case of settlements with ˂2000 equivalent inhabitants, the directive only specifies that an appropriate treatment is required, without establishing the limits with respect to the concentration of dissolved organic matter or suspended solids in the treated effluent.

For the above reasons, numerous studies have been undertaken in order to improve the degree of knowledge with respect to the purification of wastewater in small population settlements through the use of different technologies and to ensure that the decision-making process can be carried out with certain guarantees when it comes to their application.

It should also be noted that countries such as France, Poland, the UK, and Austria now have legislation that is specifically aimed at the treatment of the wastewater that is generated in small communities that does include discharge limits [11]. In the case of Spain, among the objectives of the National Plan for Wastewater Treatment, Sanitation, Efficiency, Savings, and Reuse [15] is the promotion of low energy consumption treatment technologies for small population settlements.

The Canary Islands, which are situated facing the northwest coast of Africa, have a distinctive climatology, and their morphological and geological characteristics, which are shaped principally by their volcanic origin, condition the access to and the ways of exploiting the islands’ natural water resources. In addition, a high population density, with its associated economic activities, has resulted in a significant water deficit that is particularly noticeable on the more populated islands. The orographic complexity of the islands, and the circumstance of a scattered population, add to the difficulty of ensuring the presence of adequate wastewater treatment systems throughout the entirety of the islands [16].

In view of the above issues, the aim of reducing energy, economic, and environmental costs suggests the need to search for sustainable alternatives for the treatment of wastewater in small communities. Any proposed solution for the implementation of such alternatives should take into consideration the following aspects [17]:A system that is environmentally integrated;An adequate effluent quality, whether the treated water is to be discharged or reused;A system that is able to adapt to fluctuations in the flow rate and the pollutant load;Minimum or null electric energy consumption;Simple and low cost maintenance;Minimum sludge production.

There are a number of valid purification systems for small populations that are based on low energy consumption technologies that use natural processes as the purifying element. For small communities and remote areas, nature-based solutions (NBS) are gaining popularity [18]. These include green filters, artificial wetlands, and stabilization ponds, and have shown good results when specific analyses have been undertaken of the operating requirements and the resources that are in place for appropriate sizing and follow-up.

The systems of this type are attracting particular attention at the present time as an alternative technology for the treatment of the general wastewater that is generated in small, rural, and/or isolated settlements given, as well as many other advantages, the good pollutant removal efficiencies that can be achieved with them.

It should be highlighted that the use of such types of system are in line with the objectives of the circular economy action plans of the European Union, as they promote the use of regenerated water with a suitable nutrient content for irrigation [8], as well as the use of the plant biomass that is generated in the wetlands.

One such system, which has been used over the last 60 years worldwide, is the constructed wetland (CW) wastewater treatment plant. Of these, hybrid constructed wetland systems (HCWS), particularly the vertical-flow–horizontal-flow type, are specifically recommended as they ensure high efficiency in wastewater treatment with a relatively low energy demand [19].

The published studies indicate the suitability of NBS systems [3], and in particular CWs, as a means to obtain an effluent that can be used in agricultural irrigation with good quality levels in terms of conductivity, chemical oxygen demand (COD), total nitrogen (total N), K, Ca, and Mg [20]. These wetland systems are sensitive to the type of gravel that is used [21], but are robust against climatic changes, such as the increased frequency of storms [22].

Studies in the literature [3] have shown that HCW systems can generate treated wastewater with a sufficient quality for its reuse in agricultural irrigation, according to EU regulations [23].

Nevertheless, our literature review has revealed very few studies on HCWs systems that have been operating for long periods, as well as little information on systems of this type that have been subjected to significantly higher hydraulic and pollutant loads than were contemplated in their design.

On this, we decided to conduct this research in order to make up for the shortfall of information in this field.

The objective and the novelty of this work is to evaluate, for a 6-year period (January 2014–December 2019), the performance of an HCW system with null electric energy costs in the treatment of the wastewater that is generated by the population settlements that are situated in Caldera de Tirajana (Santa Lucía, Gran Canaria, Spain).

Since its inauguration in 2008, the system has been working under hydraulic and pollutant loads that are considerably higher than those that were contemplated in its initial design. The flow and the pollutant loads also undergo wide variations, which is typically the case when dealing with small urban agglomerations.

This study involves an evaluation of the removal efficiencies, with respect to the concentrations of several analyzed parameters, both over time (as relatively few studies have been published on systems with over ten years of operation) and in terms of the different elements of the system.

An evaluation is also made of the influence of rainfall on the influent water quality and of temperature on the water quality at each sampling point.

Other possible aspects are also identified that need to be taken into account when evaluating the system efficiency, including meteorological variables and possible problems in the system management.

The results that have been obtained are compared with those from a previous study on the same installation [24], as well as with the results that have been published in the literature for other similar systems.

The system, which has been in service since 2008, was installed by the Southeast Community of Municipalities of Gran Canaria within the framework of the DEPURANAT project [5], was co-funded through the INTERREG III B Atlantic area Community Initiative, and was in cooperation with the New Water Technologies Centre (Centro de Nuevas Tecnologías del Agua—CENTA) in Andalusia and the Canary Islands Institute of Technology (Instituto Tecnológico de Canarias—ITC).

## 2. Materials and Methods

### 2.1. Description of the System

Figure 1 shows the general layout of the HCWS in Santa Lucía, its different stages, and the sampling points used for this study. The treatment process comprises the following [5]:Inlet structure: The wastewater from a non-separated sewer network reaches the system through a collector pipe. The inlet structure includes a rainwater gate and a spillway.Bar screen (A): A coarse, manually-cleaned wastewater screen composed of bars 2 cm apart.Septic tank constructed in situ (B): With 2 chambers and a total capacity of about 70 m^3^.Prefabricated Imhoff tank (C): With a capacity of 15 m^3^ and manufactured by Shaler (ref. CHC-IMH).Distribution basin with sump area for intermittent wetland discharge (D): Capacity of 6 m^3^ with safety overflow, one wastewater inlet from the Imhoff tank, and three outlet connections: two to feed each of the vertical-flow constructed wetlands (VFCWs) and one bypass to the horizontal-flow constructed wetland (HFCW). This system allows intermittent VFCW discharge without the need for feed pumps.Two VFCWs (E and F): These are identified as right (R) and left (L), respectively. The surface area of the one on the right (E: RVFCW) is 150 m^2^ and that of the one on the left (F: LVFCW) is 170 m^2^. The filter substrate is composed of a 20 cm layer of 20–32 mm gravel, where drainage pipes are embedded, which connect to aeration chimneys, and a second 80 cm surface layer of 6–12 mm gravel, creating a total substrate depth of 1 m. The VFCWs are used in alternating periods of approximately one month, with one in operation while the other is at rest.One HFCW (G): The HFCW has an effective treatment surface area of 330 m^2^ (24.5 m long by 13.5 m wide) and is planted with *Typha latifolia*, which is a plant species that is harvested twice a year from Caldera de Taburiente and used by local artisans in their handicrafts, as bedding for livestock, and mixed into manure for its reuse as fertilizer. As several studies have reported [9,25,26,27,28], vegetation contributes to the system’s performance, and is therefore an important component of wetlands’ treatment systems.

The waters from the VFCW are sent to the head of the HFCW. Discharge takes place through a distribution pipe positioned on the surface of the gravel. The waters run horizontally through a porous medium composed of different sized gravel (40–80 cm in the head and final drainage areas, and 4–12 mm in the central area) before reaching the outlet basin.

Storage lagoon (H): The effluent is taken to a 62.5 m^3^ storage pond from where it is distributed for the irrigation of olive trees or, occasionally, discharged into a dry canyon.

In these systems, contaminant removal takes place through adsorption, microbial degradation (elimination of heterotrophic organic compounds, ammonium assimilation, nitrification, and denitrification), and absorption processes on the part of the plants [29]. A configuration that combines different types of constructed wetland is appropriate in order to balance the strengths and weaknesses of each one [30], resulting in increased treatment efficiency, especially in the case of nitrogen. In the latter respect, the main role of the VFCWs in these hybrid systems is to maximize nitrogen elimination through nitrification, while that of the HFCWs is denitrification [31,32]. In addition, the alternation of the VFCWs, with periods of load and rest, allows for the application of high organic loads [33].

The average registered population of the resident Santa Lucía population (the principal source of the wastewater to be treated) was 600 inhabitants in the 2008–2019 period (Appendix A). The HCWS system was sized as a pilot project for a 100 equivalent inhabitants load and a daily flow of around 12.5 m^3^. However, as a result of damage to the main pipeline that took the wastewater to a conventional WWTP that was situated 20 km away along the coast of the southeast of the island, the authorities were obligated to redirect all of the generated wastewater for treatment by the newly inaugurated HCWS. In consequence, the system has been operating under hydraulic and pollutant loads far higher than those contemplated in its original design, which may have affected the system’s overall performance.

As can be observed in Appendix A, the registered population has fallen by just 2.5% since the system was put into operation, reaching a maximum of 626 in 2012 (Source: Instituto Canario de Estadística (ISTAC)-Instituto Nacional de Estadística (INE)).

### 2.2. Flow Rate Measurements

A number of measurement campaigns have been undertaken since the inauguration of the HCWS in July of 2008, with the aim of characterizing the plant’s input flow rate. For this purpose, three SIEMENS SITRANS F M MAG 8000 DN50 electromagnetic flowmeters were used. Two were installed in the feed lines to each VFCW, while the third was used to measure the flow that might be diverted to the HFCW in the event of an overflow of the distribution basin.

However, these flow rate measurements were not undertaken on a regular basis over the course of the operating period of the system, with 2015 being the last year of available data.

### 2.3. Meteorological Variables

The meteorological variables were obtained from the Territorial Delegation in the Canary Islands of Spain’s State Meteorological Agency, which is known as AEMET (Agencia Estatal de Meteorología). The data were taken from the nearest weather station (at a distance of 450 m), C636K, named “Santa Lucía Tirajana—Casco”, and situated at an altitude of 690 m above sea level with the following coordinates: 27°54′35″ N and 15°32′40″ W.

The data obtained were the mean monthly temperature (°C) and the mean daily rainfall (L/m^2^), from 07:00 of the day for which the data were recorded to 07:00 of the following day.

### 2.4. Sampling Plan and Parameters Analysed

The six sampling points are shown in Figure 1 and were as follows:Wastewater influent;Septic tank effluent;Imhoff tank effluent;LVFCW effluent;RVFCW effluent;HFCW effluent.

Note: The left and right VFCWs operate in alternation, and so the samples were taken from the VFCW that was in operation at the time of sampling.

In the summer of 2010, an hourly sampling of the influent and the effluent of the HCWS of Santa Lucía was undertaken on both workdays and weekends in order to determine whether there were any possible hourly and/or daily variations in the influent. It was concluded from that study that the weekday point sampling that was undertaken was representative of the overall weekday sample values in terms of both concentration and biodegradability. However, the weekend sample values differed from those of the workdays in terms of a lower influent concentrate and a less favorable 5-day biochemical oxygen demand (BOD_5_)/COD ratio, with respect to biodegradability [24].

For this reason, spot samples were taken between January 2014 and December 2019, approximately every two weeks on workdays, and in the morning (between 09:00 and 12:00), making a total of 153 sampling campaigns and the collection of more than 900 wastewater samples.

The parameters that were analyzed were BOD_5_, COD, total suspended solids (TSS), total N, and ammonium (NH_4_).

All of the parameters were measured following the standard methods for the examination of water and wastewater of the American Public Health Association [34]. The BOD_5_ was measured using OxiTop^®^ manometers (WTW). The digestion step for COD and total N was performed using a Hach digester (LT 200). The COD (50–300 mg/L or 100–2000 mg/L range), NH_4_ (2.5–60 mg/L or 60–167 mg/L range), and total N (20–100 mg/L range) were diluted, if necessary, and were analyzed using appropriate and certified Hach Lange cuvette LCK tests and a visible spectrophotometer (Hach, DR 3900). The TSS was determined by filtration and gravimetry.

### 2.5. Statistical Analysis of the Data

The collected data were tabulated in an Excel spreadsheet and the following statistical analyses were performed using the Jamovi 1.6.23 (www.jamovi.org) software programme:

Pearson’s r test [35,36]
Effect of daily rainfall on influent quality;Effect of the mean monthly temperature on water quality at each sampling point, since temperature influences the removal of pollutants in constructed wetlands [37];Effect of the passage of time on HFCW effluent quality.


Student’s *t*-test


This work is based on the hypothesis that, given the considerably large amount of data, it can be assumed that these follow a normal distribution of the mean. In order to determine whether there were statistically significant s in the effluent water quality of the alternately operating LVFCW and RVFCW, the data series were subjected to the Student’s *t*-test.

## 3. Results and Discussion

### 3.1. Treated Flow

The mean daily flow of the installation and its standard deviation had to be taken as an estimated reference based on the data that were collected in the previous years and are shown in Table 1.

It should be noted here that in 2008 and 2009, adjustments had to be made to the system due to hydraulic overloading, in which various incidences and obstructions took place in the distribution basin (D: see Figure 1), which, on some occasions, caused a decrease in the untreated flows entering the system, which may explain the low minimum flow rate values of these years.

It can also be observed that more marked episodes of maximum flow corresponding to the discharges from the municipal swimming pool can be detected in the years with a higher number of data records (e.g., 2012 and 2013).

Despite the above points, the mean daily flow values for all of the years are between 26.8 and 35.2 m^3^/day. For the purposes of the calculation of the pollutant load that the system supports, the average of the mean daily flow corresponding to 2010 and 2013 was taken as the benchmark, as these years have the highest number of data records and, therefore, they better represent the flow that was treated by the system, which was 34.25 ± 18.6 m^3^/day of wastewater.

### 3.2. Meteorological Variables

Using the data that were provided by AEMET, the mean values of the monthly temperature and the daily rainfall in the zone during the study period are shown in Appendix A.

An analysis of the data shows that the coldest months are from December to February, with values of 12–15 °C. The warmest months are July and August, with values of 25–30 °C. The temperature difference between the winter and the summer is no greater than 15 °C, and on no occasion were there frosts that might have compromised the functioning of the system.

With respect to the daily rainfall values, the rains are occasional and more frequent in the months from October to March. It should also be noted that, in the period that was evaluated, the mean number of days with rain over the course of the year was only fourteen. In six of the evaluated years, there were only twelve days with daily rainfall values that were higher than 30 L/m^2^, and only two days with values above 70 L/m^2^. The driest year was 2017, with a total rainfall of 85 L/m^2^, and the wettest year was 2019, with a total rainfall of 365 L/m^2^. As can be seen, this is an arid environment with sparse, irregular rainfall and very occasional downpours.

### 3.3. Characterization of the Water at the Different Sampling Points

The results that were obtained over the course of the study period for the selected parameters were analyzed, with a special emphasis on those that are most commonly used as wastewater pollutant indicators: BOD_5_, COD, and TSS.

#### 3.3.1. Influent of the Santa Lucía HCWS

Table 2 shows a summary of the quality of the wastewater that was to be treated for the period 2014–2019, based on spot sampling approximately every two weeks. As can be observed, the influent that was to be treated by the HCWS of Santa Lucía (sampling point 1) can be considered to be of a very strong concentration, as well as being biodegradable [38].

The high concentration of the influent in the HCWS of Santa Lucia can be attributed to the patterns of saving and the efficient use of water of the rural population of Gran Canaria, which has traditionally suffered scarcities of water resources, resulting in a low water consumption per capita. The high concentration and biodegradability of the influent can also be attributed to family activities involving the processing of agricultural and livestock products, which are typical activities in the rural areas.

Stable BOD_5_, COD, and TSS values can also be seen from the beginning of the operation of the system, with the mean values of all of the parameters being very similar to those that were obtained in a previous study on the same installation [24].

The evolution of the BOD_5_, the COD, the TSS, the total N, and the NH_4_ values in the influent of the HCWS of Santa Lucía throughout the study period are shown in Appendix A, respectively.

In all of the cases, the trend lines indicate stable BOD_5_, COD, TSS, total N, and NH_4_ values in the influent during the study period.

On the basis of the mean BOD_5_ value and the mean system input flow, the mean pollutant load that was to be treated is 25.6 kg BOD_5_/day, which is a similar value to that found in a previous study on the same installation [24].

With the available data, an analysis was undertaken as to whether the days of rainfall affected the influent characteristics, either in the sense of it being more diluted or, in contrast, due to the transport of the accumulated solids in the sewer network. The main limiting factor in this analysis was that the sampling days did not necessarily coincide with the days of rainfall or the days following rainfall, with the consequence that the results that were obtained cannot be considered to be conclusive. Nonetheless, a Pearson’s r test was performed, with the results showing no association between the influent quality and the rainfall (all *p*-values < 0.05). The same result was found when studying the influence of the mean monthly temperature on the influent quality.

#### 3.3.2. Primary Treatment Effluent

Table 3 displays the results of the analysis of the quality of the water after the different primary treatment stages: the septic tank (sampling point 2) and the prefabricated Imhoff tank (sampling point 3), which are arranged in series. Table 3 shows the mean, the maximum, and the minimum values, the standard deviation, and the number of samples that were taken for the different parameters studied here, as well as the septic tank effluent values that were obtained from a previous study on the same installation [24].

The septic tank ensures a hydraulic retention time of approximately one day, allowing the homogenization of the water entering the Imhoff tank and lowering the fluctuations in the concentrations of the parameters. The analysis of the biodegradability at these two sampling points showed similar values to those that were obtained at sampling point 1 (0.67), meaning that the primary treatment is able to treat homogenous and biodegradable wastewaters.

With respect to the septic tank effluent, the mean values of all of the parameters are, again, similar to those that were obtained in a previous study on the same installation [24], which did not include Imhoff tank effluent values.

Appendix A, show an overall perspective of the evolution of the BOD_5_, COD, TSS, total N, and NH_4_ values after primary treatment (septic tank and Imhoff tank) during the study period, respectively.

As in the case of the influent of the system, the trend lines that were obtained for the BOD_5_, the COD, the TSS, the total N, and the NH_4_ values in the primary treatment effluent also indicate stability after this primary treatment stage.

The Pearson’s r test was applied to the values that were obtained for the different parameters in the primary treatment effluent in order to determine whether there was a correlation between these and the mean monthly temperature for the same time period.

With a 95% confidence level, a correlation was found between the BOD_5_ and the temperature, and between the COD and the temperature, with respective Pearson r values of −0.154 and −0.197. This indicates, in both cases, a low and negative correlation; as the temperature rises, the BOD_5_ and COD values fall (Appendix A), as would be expected given that a higher temperature provides more optimal conditions for the biological processes and the biological activity of the microorganisms [39], which lower the organic matter concentrations.

No correlation was found between the temperature and the other parameters that were analyzed.

#### 3.3.3. Secondary Treatment Effluent

Table 4 displays the results of the analysis of the quality of the effluent from the secondary treatment, comprising two VFCWs (left and right) and one HFCW.

Table 4 shows the mean, the maximum, and the minimum values, the standard deviation, and the number of samples that were taken for the different parameters studied in the effluents of both of the VFCWs (sampling points 4 and 5) and the HFCW (sampling point 6).

It should be noted, in this case, that the values of the HFCW influent are determined by the mean of the values of the effluents of the two VFCWs.

The results that are shown in Table 4 do not differ significantly from those that were reported in a previous study on the same installation [24], with the values for the right and the left VFCWs of 146 and 127 mg/L of BOD_5_, 332 and 298 mg/L of COD, 49 and 50 mg/L of TSS, 75 and 67 mg/L of total N, and 57 and 42 mg/L of NH_4_, respectively.

In the aforementioned study [24], the HFCW effluent results are similar to those that have been found here, with 49 mg/L of BOD_5_, 138 mg/L of COD, 8.5 mg/L of TSS, 45 mg/L of total N, and 48 mg/L of NH_4_, respectively.

The results are also comparable and similar to the data that were provided by studies such as [40,41,42,43,44], with values ranging for COD from 80.83 to 113.82 mg/L, for BOD_5_ from 21.27 to 57.32 mg/L, for TSS from 25.89 to 64.63 mg/L, for NT from 17.17 to 45.72 mg/L, and for NH_4_ from 30.11 to 34.23 mg/L, respectively.

Vertical-flow constructed wetlands, VFCWs

(a) VFCW effluent (considered jointly)

Appendix A, show the sequence of the values that were obtained for BOD_5_, COD, TSS, total N, and NH_4_ in the effluent of the different VFCWs, respectively. It can be seen how, in the periods of operation of the RVFCW, the LVFCW remains inactive and vice versa.

It can be observed in Appendix A that the BOD_5_ values for both of the wetlands tend to decrease over time, and that the decrease is more marked for the LVFCW.

While the COD effluent values of the two VFCWs (Appendix A) range around 300 mg/L throughout the study period, a slight tendency can be seen for a decrease in the case of the LVFCW and for an increase in the case of the RVFCW.

Besides the COD values, something similar occurs with the TSS in the VFCWs (Appendix A). The values range around 50 mg/L, with a slight decreasing tendency in the case of the LVFCW and slight increasing tendency in the case of the RVFCW.

In Appendix A, it can be observed that the total N values for both of the wetlands tend to increase over time, with a more marked increase for the RVFCW.

As can be seen in Appendix A, both of the NH_4_ trend lines are parallel, and they show a slight increase over time, with the RVFCW trend line being above that of the LVFCW trend line.

The obtained data suggest that there may be some operational differences between the two VFCWs. This issue is analyzed and discussed in the following subsection.

(b) Effluents of the left and right VFCWs

With respect to the functioning of the VFCWs, it should be noted that they do not operate on the same days, on the same number of days, or with the same pollutant loads, therefore, the quality of the wastewater influent of each VFCW is not necessarily the same.

A Student’s *t* test was performed in order to determine any possible differences in the concentrations of the parameters that have been studied in the effluent of the alternately operating LVFCW and RVFCW. Such differences in concentrations could be indicative of different operation and, therefore, different efficiencies. No significant differences were found, except in the case of NH_4_ (*p*-value = 0.004 < 0.05). The different trends that were observed in the BOD_5_, COD, TSS, and total N values are not such that it can be affirmed that the data series corresponding to the LVFCW and the RVFCW are significantly different.

As can be clearly seen in Appendix A, the NH_4_ trend lines of the two VFCWs are parallel and show a slight increase over the course of the study period, with the RVFCW trend line being above that of the LVFCW. This behavior may have been taking place since the inauguration of the system, as lower NH_4_ concentration values were reported in the previous study that was undertaken on the same installation (RVFCW = 57 mg/L and LVFCW = 42 mg/L) [24].

As the passage of time could be a key factor in explaining the difference that was observed for NH_4_. Appendix A shows the period of activity of each VFCW, as well as their respective total days of operation.

It can be seen how the RVFCW was in operation more often, and, hence, active for more days (1220 days) than the LVFCW (1062 days).

Ammonium removal by nitrification is enhanced in the presence of oxygen [45]. The longer operating period of the RVFCW may be leading to a greater accumulation of sludge and, in turn, to more anaerobic conditions, resulting in higher ammonium concentrations in the RVFCW effluent than in the LVFCW effluent.

A higher temperature also favors NH_4_ removal. The fact that the RVFCW is partially shaded by a large nearby eucalyptus tree may, therefore, also be contributing to the lower NH_4_ removal efficiency of the RVFCW compared to the LVFCW.

Finally, the smaller size of the RVFCW (150 m^2^), compared to the LVFCW (170 m^2^), may also be contributing to the differences in the results, at least with respect to the NH_4_.

(c) Effect of temperature on the effluent of the VFCWs

The Pearson’s r test was performed in order to determine whether there was any correlation between the overall values of the VFCW effluent parameters (without distinguishing between the RVFCW and the LVFCW) and the temperature. A correlation was found between the BOD_5_, the COD, and the TSS values and the temperature (Appendix A), with the relationship in all cases being weak and negative (Pearson r values of −0.227, −0.271, and −0.205, respectively).

In other words, as the temperature rises, lower concentrations of BOD_5_, COD, and TSS are found in the effluent of the VFCWs, indicating that higher temperatures have a positive impact on the quality of the treated wastewater.

As previously mentioned, most of the biological processes are favored by a higher temperature [9,18,22,39,46], which explains why this lowers the concentrations of organic matter in this case the BOD_5_ and the COD.

With respect to the decrease in the TSS as the temperature increases, this could be attributable to the large contribution of organic matter in its composition, the degradation of which is favored by higher temperatures. In addition, higher temperatures favor the solubility of particular substances that are present in the water, which contributes to lowering the TSS values. Finally, higher temperatures also favor the growth of *Typha* spp. and root development in the wetlands, which may also be contributing to particle retention.

Annual mean temperatures range between 15 and 25 °C and exceed 30 °C during the summer months, thus providing optimal conditions for many biological processes and the biological activity of microorganisms [21].

*Typha* spp. and root development in the wetlands, which may also be contributing to particle retention.

Horizontal-flow constructed wetland, HFCW

(a) HFCW effluent

Table 4 shows the results with respect to the quality of the HFCW effluent. Appendix A, represent the concentrations that have been found over the course of the study period for BOD_5_, COD, TSS, total N, and NH_4_, respectively.

An increasing trend over time can be observed for the BOD_5_, the COD, and the NH_4_ values, as well as a tendency to lower the TSS and the total N values.

The Pearson r test only found a correlation between the TSS and time (*p*-value: 0.037 and r = −0.18), which could be attributable to the natural evolution of the HFCW and its associated phenomena (compaction, silting, etc.,), contributing to the improved ‘filtering’ of solids, as other studies have found [47].

(b) Effect of temperature on HFCW effluent

The analysis that was undertaken in order to determine the existence, or otherwise, of a relationship between the temperature and the study parameter values in the HFCW effluent showed, with a 95% confidence level, no correlation in all cases.

### 3.4. System Removal Efficiencies

Table 5 shows the mean removal efficiencies (%) that were obtained on the basis of the mean values of the parameters that were measured at each of the sampling points of the study, comparing the water quality of the influent (C_influent_) and the effluent (C_effluent_) of each of the treatment stages [19], as follows:% = 100 × (1 − C_effluent_/C_influent_) 

The highest removal efficiency in the primary treatment stage was obtained through the septic tank, with the Imhoff tank functioning as a complementary treatment, but with a relatively low impact, especially in terms of the nutrients. The septic tank removal efficiency ranged between 11% and 54% for NH_4_ and TSS, respectively. In the previous study [24], the septic tank removal efficiencies (Appendix A) were 36% for BOD_5_, 34% for COD, 38% for TSS, 18% for total N, and 13% for NH_4_.

With respect to the primary treatment stage as a whole (Figure 1), it can be seen how the removal efficiencies for the primary treatment were 42% for BOD_5_, 43% for COD, 65% for TSS, 15% for total N, and 8% for NH_4_.

As for the secondary treatment stage, a higher removal efficiency was obtained for the VFCWs (44%), compared to the HFCW (36%).

In the case of the VFCWs, the removal efficiencies ranged between 22% and 71% for NH_4_ and TSS, respectively. The corresponding values for the HFCW were 8% for NH_4_ and 70% for TSS. The BOD_5_, the COD, and the TSS removal efficiencies were above 50% for both the VFCWs and the HFCW.

The removal efficiencies of the VFCWs are lower than those that were found in a previous study (Appendix A) on the same installation [24]; however, the removal efficiencies of the HFCW have increased. This may be due to the HFCW having to support a higher pollutant load as the result of poorer VFCW performance due to lack of maintenance, excessively long operating periods, etc.

According to [48], shorter alternating operating periods of VFCWs allow for the better control of biomass growth, the maintenance of the aerobic conditions in the filter bed, and the mineralization of the organic deposits that accumulate on the surface of the bed. In one of the studies that was consulted, alternating periods of 3.5 days were used [33].

As can be seen in Figure 2, removal efficiencies of above 80% were obtained in the secondary treatment stage as a whole for three of the five parameters that were studied, with the TSS recording the highest value (91%), followed by the BOD_5_ (86%), the COD (81%), the total N (38%), and the NH_4_ (29%).

The values have slightly decreased when compared (Appendix A) with the previous study [24], where BOD_5_, COD, TSS, total N, and NH_4_ were 88%, 81%, 97%, 54%, and 40%, respectively.

Figure 3 shows the overall performance of the system over the course of the study period. The BOD_5_, the COD, and the TSS overall removal efficiency values are more than satisfactory, with values of 92% for BOD_5_, 89% for COD, and 97% for TSS, confirming that the removal of the organic matter and the suspended solids is optimal for a system of its characteristics. The BOD_5_, the COD, and the TSS values are practically the same as those that were obtained in a previous study [24], while the total N and NH_4_ removal efficiencies have decreased by around 25% (Appendix A).

In a review of the results of the pollutant removal efficiencies that were obtained in different studies that were conducted between 2000 and 2013 on wetland wastewater treatment systems in tropical and subtropical regions [46], the following ranges were obtained for hybrid systems that were similar to the one that has been considered in the present study: 52–92.26% for BOD_5_; 71.66–97.72% for COD; 79–97.49% for TSS; 63.41–91.33% for total N; 62.50–91.20% for NH_4_.

The only parameters of the system that were considered in the present study that are outside these ranges are the total N and NH_4_, as was also the case for the values that were obtained in a previous study on the same installation [24]; however, given that the treated water is used for olive tree irrigation purposes, this presence of nutrients can be considered to be an advantage rather than a drawback.

Another more recent review [49] has reported removal efficiencies of 84–96% for BOD_5_, 74–95% for COD, 85–99% for TSS, 26–78% for total N, and 19–91% for NH_4_. In this case, all of the values that were obtained in the present study are within these ranges.

Table 6 shows the removal efficiencies that have been reported in other studies on the treatment of urban wastewater using systems that combine VCFWs and HFCWs in regions of mild temperatures. More step-by-step details for some of these are given in Appendix A.

In view of the above information, it can be affirmed that the overall removal of the organic matter and the suspended solids was optimal in comparison with the results that were obtained in similar installations.

Nonetheless, low nutrient removal efficiencies have again been observed, especially for nitrogen compounds. This may be due to the fact that the system design was based on the use of BOD_5_ as the parameter of choice and not nitrogen [49]. The low performance may be correctable by maintaining high levels of oxygenation in the VFCWs [33] and/or ensuring that the water that is to be treated in the HFCW has sufficient organic matter in order to enable heterotrophic bacteria to carry out the denitrification reaction [25]. With respect to the latter, a TOC (total organic carbon)/TN (total nitrogen) ratio of between 2.5 and 5.0 has been proposed [52].

The results that have been presented in this paper are of significant importance, not only demonstrating the feasibility of HCWS as a promising alternative for the reliable and efficient treatment of wastewater, but also for the additional reuse of reclaimed water for irrigation in small population settlements.

It is an interesting study that provides valuable information from over six years’ worth of data, with the novel feature that the HCW system has been in continuous operation since 2008 and has been subjected to a hydraulic and organic load that was three to four times higher than contemplated in its design.

Despite this, the system continues to function correctly, treating one equivalent inhabitant load in approximately one square meter. This result can contribute to promoting interest in this technology for the treatment and reuse of wastewater in small populations. In terms of future studies and/or designs of similar systems, the practical applicability of this study lies in the possible reduction in sizing requirements (which is a major limitation of its application [53]) and, consequently, the associated costs.

## 4. Conclusions

This study indicates that the performance of an HCW system was, in general, stable over the course of its 11 years of operation, between 2008 and 2019, despite it being subjected to higher than planned for hydraulic and pollutant loads. The results in terms of m^2^ per inhabitant equivalent ratios of both of the VFCWs and the HFCW, compared to the ratios that were suggested by diverse authors, may be indicative of an oversizing of the system during its design stage. That is to say, it may be possible to reduce the surface area that is occupied by such wastewater treatment systems given the results that have been presented in this study. The results that have been obtained confirm that the natural systems of the type contemplated in the present study constitute a proven, robust, and long-lasting solution for the treatment of the wastewater of small population settlements, especially in rural and/or isolated areas with stable mean temperatures and mild winters. In addition to an appropriate environmental integration, other advantages of such systems include their low operating and maintenance costs, a null electric energy cost, and the in situ reuse of the treated wastewater. Finally, it should be highlighted that systems of this type are also able to remove emerging pollutants, including pharmaceutical waste, as shown in a study that was carried out at the same installation [54], but also personal care products, endocrine-disrupting compounds, etc. [55,56].

## Data Availability

Not applicable.

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
