# Peer review of "Long-Term Performance of a Hybrid-Flow Constructed Wetlands System for Urban Wastewater Treatment in Caldera de Tirajana (Santa Lucía, Gran Canaria, Spain)"

_ijerph, 2022, doi:10.3390/ijerph192214871_

Round 1

Reviewer 1 Report

This research provided a study about the performance of wastewater treatment plants formed from vertical and horizontal flow-constructed wetlands to treat urban wastewater.   

Here are my comments:

My recommendations:

  1. I recommend introducing some results in the Abstract section (removal efficiency, etc.).
  2. The Introduction section must be improved with more literature studies, and new references must be included (especially from 2022). You need approx. 10 bibliographic references in the field in the last three years.
  3. Please specify COD, NH4, and total N content standards, lines 248 and 250.
  4.  Please revise the standard deviation for values presented in Table 2 (414 mg/l for TSS, which I think is a mistake, maybe 41.4 mg/l). Also, the values showed in Table 4 (TSS – 47 ±47 mg/L). The uncertainty of the method is 50%; something is wrong.
  5. Please give more bibliographic indices in table 6 for a comparation of the obtained results.
  6. In L'253-266 the authors wrote about statistical processing (Pearson’s r test and student’s test). However, I cannot find any helpful information in the results section. 

Reviewer 2 Report

The manuscript “Long-term performance of a hybrid vertical and horizontal flow constructed wetlands system for urban wastewater treatment” lacks novelty. The characterization of the wastewater was based on single samples. To obtain a better performance of the treatment efficiency of the wastewatwer treatment plant in long-term, grap sampling must be carried out with respect to the flow rate in different time intervals on 24 h. My biggest concern is that the article has 20 Figures and 7 Tables which are excessive for a scientific article. Authors should better organize their information and results before submitting to the journal. Graphs lack a title on the axes of x and y.

Reviewer 3 Report

COMMENTS FOR THE AUTHOR:

Manuscript entitled "Long-term performance of a hybrid vertical and horizontal flow constructed wetlands system for urban wastewater treatment" submitted by Gilberto M. Martel-Rodríguez, Vanessa Millán-Gabet, Carlos A. Mendieta-Pino, Eva García-Romero and José R. Sánchez-Ramírez, can be accepted for publishing in the International Journal of Environmental Research and Public Health, after minor revision. 

In this real case study, the evaluation of a natural wastewater treatment system which has been in continuous operation for over 11 years in municipality of Santa Lucía, Gran Canaria, Spain was done. The system, originally designed to treat a flow rate of 12.5 m3/day with a population equivalent load of 100, has been operating since July of 2008 with a flow rate of almost 35 m3/day and a population equivalent load 25 of 400. Also, the mean total removal efficiencies during the study period (2014-2019) are optimal for a system of these characteristics: 92% for 5-day biochemical oxygen demand (BOD5), 89% for chemical oxygen demand (COD), and 97% for total suspended solids (TSS). The manuscript presents interesting results, which are relatively well organized and systematized, but the novelty and practical applicability of this study should be highlighted more. Also, the English needs to be improved through the manuscript because in some parts of the article the discussion becomes very unclear. In my opinion, this manuscript should be published in your Journal after minor revision.

Here is a list of my general comments:

·        The novelty and practical applicability of this study should be highlighted more.

·        Newer references should be included in the introduction and the discussion part. From 34 references, only 10 are from last five years.

·        It is necessary to improve the English through the manuscript

Specific comments:

o   Line 292: In the sentence: Mean monthly temperature (°C) and mean daily rainfall (l/m2)” place 2 in the superscript (l/m2).

o   Line 366: In the sentence: “which is to say the primary treatment…” it is more understandable to write: “meaning that the primary treatment…” or “which means that the primary treatment…”.

o   Line 371: Put respectively” in the end of the sentence.

o   Line 413, line 420: Put respectively” in the end of the sentence.

o   Line 432: Instead how” please write that”.

o   Line 433: In the sentence: with this decrease more marked…” it is more understandable to write: and the decrease is more marked…”.

o   Line 437: In the sentence: As with the COD values, something similar occurs…” please write Beside the COD values, something similar occurs…”.

o   Line 440: Please rewrite the sentence: All of the above suggested there may be some operational difference between the two VFCWs, a question which is considered in the following subsection. ”.

o   Line 450: In the sentence: Any such differences in concentrations…” please write any” or such”, not both words.

o   Line 452: Please delete: That is to say,”.

o   Line 452: The different trends observed in the BOD5, COD and TSS values are not such that it can be affirmed that the data series corresponding to the LVFCW and the RVFCW are significantly different. ”.

o   Line 636: In the sentence: The wastewater to be treated…” please write The wastewater …”.

Reviewer 4 Report

The aim of the work was study is to evaluate the performance of HCW (vertical + horizontal beds) with null energy cost (?) in the treatment of the wastewater from Caldera de Tirajana (Santa Lucía, Gran Canaria, Spain). It is an interesting study with data of more than six years and may be better analysed to reinforce the interest of this technology for the treatment and reuse of wastewater in small populations in increasing scenarios of prolonged droughts and water crises.

Tte title should be changed since it is a case study. It should be “Long-term performance of a hybrid flow constructed wetlands system for urban wastewater treatment in Caldera de Tirajana (Santa Lucía, Gran Canaria, Spain)”

Comments:

1) Introduction, p. 2, L. 47-48, p. 3, “…The safe reuse of treated wastewater is one of the non-conventional solutions that are being considered around the world…”

The sentence should be rewritten, because treated wastewater (reclaimed water) reuse is seen as an effective and necessary alternative source of water to compensate for the water deficit associated with droughts and water crises.

2) Introduction, p. 2, after L. 50

Your study has interesting results, and for a good period of time, to conclude that the application of CW technologies can be a good alternative of nature-based solutions for, not only the treatment of wastewater, but the reuse of water in small aggregated.

Please have a look in the following studies and improved your introduction with a paragraph on reclaimed water reuse associated to CW:

DOI: https://doi.org/10.3390/w13091165

DOI: https://doi.org/10.2166/wpt.2011.050

DOI: https://doi.org/10.1016/j.ecoleng.2010.03.009

DOI: https://doi.org/10.3389/fenvs.2022.836289

3) Introduction, p. 3, L. 116-120

The use of nature-based solutions, namely technologies with algae, CW and soil infiltration, will be an appropriate solution for small populations in regions of water scarcity. You focus here on the importance of these solutions for the reduction of energy consumption. In addition to this advantage, these solutions enable carbon sequestration, circulation of water, organic matter and nutrients and are sustainable within the scope of the UN SDGs.

It would be useful to develop these advantages. Check out the following works I found in a quick search on Mendley:

DOI: https://doi.org/10.3390/w13233334

DOI: https://doi.org/10.3390/w14132042

DOI: https://doi.org/10.1016/j.rser.2021.111261

DOI: https://doi.org/10.3390/su11246981

4) Introduction, p. 3, L. 129 – 130, “...The main aim of this study is to evaluate the performance of a natural wastewater treatment system (NWWTS) with null energy cost in the treatment ...”

The term " NWWTS" is not well applied here, because in reality an hybrid CW system was studied. So "hybrid constructed wetlands (HCWs)" looks better.

I'm not sure they've had zero energy costs in six years. Energy consumption has been low in the HCW operation, but not zero in the overall operation of the plant. Adjust these types of conclusions, energy consumption was low, not null, even because it was not accounted for all items involving the operation and maintenance of HCW.

Therefore, “NWWTS” should be replaced by “HCWs” or “HCWS” and “natural wastewater treatment system” should be replaced by “hybrid constructed wetlands”, in all text, figures an tables.

5) Point 3.3.1

BOD5, COD and N value are high for a raw domestic wastewater in small communities. 

There must have been contributions from small farms and family agri-food industries that could justify the high values of biodegradable organic matter (BOD5/COD = 0.67).

Elevated values of N must be associated with leaching from agricultural land (i.e. associated with the use of fertilizers).

These aspects must be discussed to better justify the observed results.

6) Figures

What are the dotted lines? Average values? It should be inserting a legend in the figures for the points, dotted lines and other symbols.

Figures 3 to 5 are not necessary. Only the explanation of the variation of concentrations as I suggested in 5). The objective of the work is to study the performance of HCWs not the primary treatment.

7) Section 3.3.2.

Those figures are important, but a similar figure with N variation should be included. N variation should be commented as you did for COD, BOD5 and TSS.

8) Section 3.3.3.

You have many information here on the removal of COD, BOD5, TS and N. Your results should be compared with similar studies obtained in similar conditions to better understand the differences and similarities.

I found several interesting studies in areas with similar clime than Canarias Islands that can be used for comparison (it is important to have comparison with systems with influence of Mediterranean clime):

DOI: https://doi.org/10.3390/su14116511

DOI: https://doi.org/10.3390/chemengineering2010003

DOI: https://doi.org/10.3390/w13172431

DOI: https://doi.org/10.3390/w13081086

DOI: https://doi.org/10.3390/en15010156

9) Section 3.4

Figure 18 is not necessary.

10) Conclusions

Conclusions are too long. Shorten the text to one paragraph. Put only the main conclusions.

Round 2

Reviewer 2 Report

Accept